# Integrating Diverse Perspectives for Managing Neighborhood Trees and Urban Ecosystem Services in Portland, OR (US)

Lorena Alves Carvalho Nascimento * and Vivek Shandas

Nohad A. Toulan School of Urban Studies and Planning, Portland State University, Portland, OR 97201, USA; vshandas@pdx.edu
* Correspondence: lorena.nascimento@pcc.edu

**Abstract:** Municipalities worldwide are increasingly recognizing the importance of urban green spaces to mitigate climate change's extreme effects and improve residents' quality of life. Even with extensive earlier research examining the distribution of tree canopy in cities, we know little about human perceptions of urban forestry and related ecosystem services. This study aims to fill this gap by examining the variations in socioeconomic indicators and public perceptions by asking *how neighborhood trees and socioeconomic indicators mediate public perceptions of ecosystem services availability.* Using Portland, Oregon (USA) as our case study, we assessed socioeconomic indicators, land cover data, and survey responses about public perceptions of neighborhood trees. Based on over 2500 survey responses, the results indicated a significant correlation among tree canopy, resident income, and sense of ownership for urban forestry. We further identified the extent to which the absence of trees amplifies environmental injustices and challenges for engaging communities with landscape management. The results suggested that Portland residents are aware of tree maintenance challenges, and the inclusion of cultural ecosystem services can better address existing environmental injustices. Our assessment of open-ended statements suggested the importance of conducting public outreach to identify specific priorities for a community-based approach to urban forestry.

**Keywords:** urban forestry; cultural ecosystem services; public survey; tree maintenance



## 1. Introduction

By 2050, the United Nations suggests that the world human population will near 10 billion, with most living in urban centers. In the United States (US), eighty percent of the population already live in urban areas, which corresponds to only 3% of national land [1]. The fast pace of urbanization and landscape change caused by humans are the major factors for transforming forests, urban and otherwise [2]. Some have argued that the transformation of urban landscapes brings an 'extinction of experience' with nature [3], which impacts the well-being, public health and empathy for natural features. The management of urban areas requires the consideration of multiple land-use possibilities for conservation of built or natural environments. Roads, buildings, urban renewal, green infrastructure and new developments compete for limited urban space. This fact requires municipalities to use strategic approaches to manage urban growth with civic services, including sewer, roadways and other gray infrastructures. Often with priories of gray infrastructure, the available locations for tree canopy is reduced and existing tree canopy is removed—a phenomenon that has been well documented across the US [4,5].

Several scholars have argued that urban tree canopy can be better managed by characterizing ecosystem services, which describes the benefits that trees and other natural landscape features provide to humans. For urban trees and tree canopy, these are often described as improvement of air quality, reduction of heat, filtering and infiltration of stormwater and a host of cultural attributes that improve the overall quality of life for residents [6,7]. Ecosystem services, in the form of tree canopy, come in three essential categories: (a) those in parks, schools, open spaces and other natural areas; (b) those on

private lands, owned by somebody other than a public agency; (c) those on streets and other public rights-of-way.

Municipal governments generally manage the urban tree canopy through a distributed model that embeds tree specialists in different public agencies or through a central division that works with other agencies. Few municipalities, if any, explicitly examine the host of ecosystem services that constitute the urban forest. We argue that urban ecosystem services as a management approach can help situate the urban forest within broader and potentially more inclusive management of natural features. The rationale behind our approach for ecosystem services lies in these three points: (a) they offer a cost-effective and functionally-based solution to major challenges facing urban areas, including rising temperatures, air pollution and flooding; (b) when properly applied, ecosystem services can support a more equitable distribution of canopy, which is currently highly centered on wealthy and white communities; (c) the consideration of ecosystem services can improve human-nature relationships, create a sense of ownership of places and provide stewardship and community engagement opportunities. Together these functions of urban trees and forests compel their careful management, though the extent to which communities view or even understand these ecosystems services remains unclear.

This study examines the role of community perspectives concerning urban canopy management by assessing the relationship between the quantity of neighborhood tree canopy, public perceptions of ecosystem services and socioeconomic indicators to support urban environmental planning. Many studies have found a positive correlation between tree canopy and residential income [8–10]. However, few examine the extent to which public opinion about planting priorities, maintenance challenges and the expectations for urban ecosystem services can be central to decision-making processes. We posit that engaging the public in myriad and creative ways in urban forestry efforts is increasingly essential. Planting and maintaining trees can promote a connection between residents and urban environmental services according to each neighborhood's needs, regarding socioeconomic, cultural and historical aspects.

*Background*

The use of urban ecosystem services (UES) in describing trees and forests is a relatively new idea [6,7]. Four categories of UES classify the services provided by the maintenance, preservation and conservation of urban forestry. First, supporting UES contributes to nutrient cycling and soil formation through tree debris and habitat for decomposers. Second, provisioning UES for urban foraging of food and natural medicine [11]. Third, regulating UES, as local climate resilience that prevents urban heat islands [12], air quality that mitigates respiratory problems [13] and stormwater catchment that controls flood and water flow [14]. Fourth, cultural UES bring socio-ecological values through self-actualization, esteem and belonging [7], connection with biodiversity and personalized ecology [15]. The literature of ecosystem services [16] is gaining popularity, with research that includes public perception evaluations and how the ecosystem services adjust to reflect community identity.

UES relies on urban forestry and green infrastructure management, including public participation in strategic planning to recognize multiple ecological needs in diverse contexts. People's involvement in managing urban forests is often heralded as necessary for ensuring a just and equitable distribution of ecosystem services. However, communities' engagement may be mediated by intersectional factors that are often not considered in planning decisions [7]. For example, older adults and children are more vulnerable to respiratory diseases and may be highly sensitive to degraded air quality; black communities have been historically excluded from desirable green areas [17]; queer identities struggle to be accepted in heteronormative nature spaces [18]; low-income residents have less canopy access [8].

Urban forestry scholars have already acknowledged the link between socioeconomic factors and access to ecosystem services in the municipalities. McPhearson et al. [19] called

attention to incorporating UES into urban planning since cities globally are rapidly increasing in population. The persistent need for environmental justice and climate resilience created a framework for using ecosystem services as planning metrics. Wilkerson et al. [7] developed an urban sociological framework to explain the intersectionality between UES planning and social demands because the variation of socioeconomic factors impacted the accessibility to green spaces. Empirical studies that measured urban accessibility based on socioeconomic indicators found geographic mismatches within vulnerable groups for the balance (demand/flow ratio) of ecosystem services of climate regulation [12], food supply and recreation [20].

While these earlier studies call out distributional inequities, we need to establish programs and policies to ensure that historically underserved communities are at the center of urban forestry programs. We argue that urban forest planning needs to acknowledge and incorporate voices from diverse communities when managing distributional equity. Moreover, we need to find effective and practical approaches for hearing voices from communities, especially about their perceptions of trees and the factors that mediate the level of saliency for expanding tree canopy in historically disinvested neighborhoods. Since the fields of forestry and more recently urban forestry, have been mainly heteronormative, white and male [18], advancing a call for expanding the participatory process to include opinions that have not traditionally heard is instrumental to ensuring canopy equity in cities.

To provide a basis for our argument, we evaluated the presence of trees and public perceptions of ecosystem services through a community survey in Portland, Oregon (OR), US. The study aimed to understand what people expect about greening strategies, tree maintenance and tree planting priorities. We specifically asked, *how do neighborhood trees and socioeconomic indicators mediate the public perceptions of ecosystem services availability?* We addressed this question through integrative analysis, involving a survey, socioeconomic indicators and spatial analysis of neighborhoods in the study area. Portland has several advantages for a study of this kind, including an inequitable distribution of trees [10]; historically unserved areas that reflect racist planning policies [21]; and late incorporation as a city in the United States, leaving with its existing tracts of large trees, even today [22]. At the same time, Portland's regional culture seems to have an explicit appreciation for urban forestry, as evidenced by establishing the first Parks Commission in 1900 and green corridors West of Willamette River planned by the Olmsted Brothers in the early development of the city services [23].

## 2. Materials and Methods

### 2.1. Study Area

Located in the Pacific Northwest region of the United States, Portland bears the earliest and most preserved forest formations in the country [24]. Once called Silicon Forest [25], over the past 20 years, Portland attracts people looking for jobs in the tech industry and individuals seeking outdoor recreation and lifestyles. The physical geography has forest fragments in hilly areas to reduce the chance of landslides, as illustrated in Figure 1. Portland's canopy cover follows geological points of interest, like rivers, wetlands and elevated formations.

Most of the city's canopy is west of the Willamette River in the Northwest (NW) and Southwest (SW) zip code sectors (Figure 1). Portland's western sectors also contain the most topographical relief for which trees protect from erosion, landslides and riverbanks [26,27]. Western sectors of the study region had extensive early conservation policies by the Parks Commission, with the Olmsted Brothers' support. This landscape architecture firm promoted urban ecology practices, such as green corridors, large urban parks, wildlife habitat and biodiversity in the early 20th Century [23].

Across the Willamette River, the eastern zip code sectors have flattened surfaces containing fewer trees than the western sectors. The eastern sectors are North (N), Northeast (NE), Southeast (SE) and East (E). The eastern sectors have a larger proportion of

industrial and commercial areas and the largest percent of the city's population [22]. The Columbia River, the second largest river in the United States [28], surrounds North and East Portland. The high susceptibility to floods resulted in a 21-mile levee system created to allow urban development in areas along the Columbia River's riverbanks. Among the 654,741 people living in Portland [29], at least 25.6% live in the East sector [22], which is growing fast in terms of ethnic and income diversity. Still, Portland has a history of racist urban planning, redlining [30,31], gentrification [21,32,33] and late incorporation of eastern neighborhoods [22] that explains the separation between low-income communities and canopy abundance.

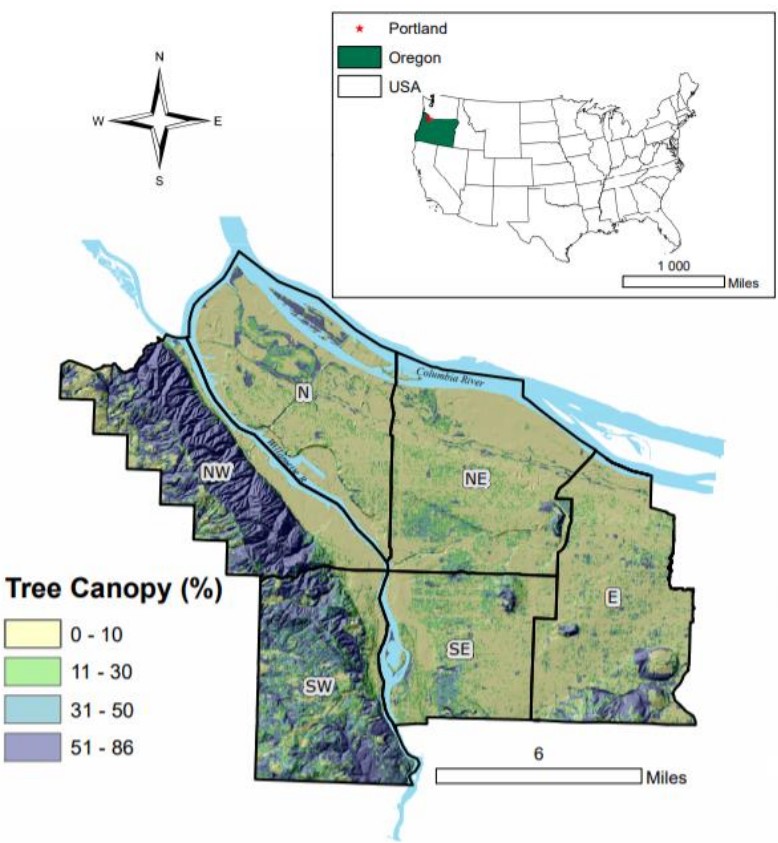

**Figure 1.** Map of Portland, Oregon, USA, with canopy distribution over the six zip code sectors: Southwest (SW), Northwest (NW), North (N), Northeast (NE), East (E) and Southeast (SE) City of Portland [34]. Oregon Spatial Data Library [35], USGS [36].

*2.2. Research Design*

We used a cross-sectional research design that applied satellite-derived measurements of the tree canopy cover, demographic analysis and assessments about the public perceptions of the urban forest. This study used three parameters that encompassed tree canopy measurements from the National Land Cover Database (NLCD), socioeconomic indicators from the US Census and a public survey about urban forestry perceptions applied in Portland. The aim was to assess the extent of the connection between people and UES following the level of satisfaction and accessibility to urban tree canopy [9]. By integrating the biophysical with survey responses, we argued that we were better able to describe how the presence or absence of neighborhood trees mediated differences in the perception of tree maintenance and tree ownership in the study area. Evaluation of public perceptions can offer insights, perhaps a first step, to understand the extent to which urban forest management can better support communities in the maintenance, ownership and accessibility to trees among different socioeconomic groups.

2.2.1. External Datasets for Tree Canopy Cover and Demographic Data

US Census data from 2017 estimates [29] regarding income, race and homeownership had two specific roles in this study. First, we wanted to note the fidelity of our survey sample. We compared the values of socioeconomic indicators from the survey and census. Second, we used census data as the parameters for socioeconomic indicators. The census data had higher representability and more complex data collection than the demographic questions from the survey. We aggregated the census data by zip code, following the delimitation of study areas from the survey.

The canopy cover percentage was obtained from the NLCD with 30 m × 30 m spatial resolution. The dataset informed the percentage of canopy per pixel [36,37]. On ArcGIS software, we used the Extract by Mask tool to mask Portland boundaries and the Tabulate Area tool to measure the canopy area per zip code. Equation (1) calculated the average canopy percentage by zip code. The shapefiles for zip code and neighborhood boundaries were from the Portland Maps web library [34]:

$$\text{Canopy cover } (\%) \; = \; \sum (\text{ZAi} \; \times \; \text{Ci}) / \text{area} \tag{1}$$

where ZA is the zip code area per canopy percentage, C is the percentage of tree canopy per pixel and area is the total zip code area.

2.2.2. Survey Data for Public Perceptions of Urban Forestry

To explore the public perceptions of UES, we used the results of an online public survey conducted between May and July of 2017. Portland is known for the neighborhood bonding and strong connection that residents have with their vicinities [25]. Therefore, we used zip codes as the determinant scale and the unit of analysis. A total of 26 zip codes were large enough to have representative samples to assess patterns of responses and small enough to provide variations in our sample. We excluded zip codes from neighborhoods with less than fifteen answers for a better variability of answers and representation.

The survey had 26 questions that explored the views of Portlanders about the quality of the local urban forest, strategies for planting programs, possibilities for tree maintenance and a socioeconomic questionnaire [38]. For this study, we combined 12 relevant questions that addressed: (a) the sense of ownership of trees; (b) the sense of maintenance for trees in public spaces; (c) the perception of UES on strategies to increase urban forestry; (d) and socioeconomic indicators (Appendix A).

The first six questions of our study encompassed the perceptions of tree ownership and maintenance [Appendix A—Questions 1–6]. Three questions about tree ownership asked the participants about: (1) the importance of trees; (2) the satisfaction with the number of trees and (3) the satisfaction with trees' health [Appendix A, Questions 1–3]. Three questions about tree maintenance inquired about (1) the maintenance of existing trees in the right-of-way; (2) maintenance of trees in the right-of-way in low-income communities and (3) planting new trees in the right-of-way [Appendix A, questions 4–6]. The participants used a Likert scale ranging between 1 and 5 to inform how much they agree or disagree with the six sentences related to tree ownership and tree maintenance. Table 1 shows the range of Likert scale values, tree ownership and maintenance questions and multi-metric evaluation.

Instead of analyzing the six questions separately, we created two multi-metric indexes: Tree Ownership Satisfaction Index (TOSI) and Tree Maintenance Satisfaction Index (TMSI). TOSI combined the three questions about tree ownership and TMSI the three questions about tree maintenance. TOSI and TMSI also used a Likert scale and we calculated them by finding the average values of the three questions of each index, as displayed in Table 1. TOSI and TMSI resemble multi-metric indexes used in the biological assessment of watersheds [39,40].

**Table 1.** Where Q is the question number, Tree Ownership Satisfaction Index (TOSI) is the tree ownership satisfaction index and Tree Maintenance Satisfaction Index (TMSI) is the tree maintenance satisfaction index.

| | Likert-Scale Values | | | | | |
|---|---|---|---|---|---|---|
| **Questions** | **Strongly Agree** | **Agree** | **Don't Know** | **Disagree** | **Strongly Disagree** | **Multi-Metric Index** |
| Q1: "Portland's trees are important to me" | | | | | | |
| Q2: "My neighborhood has enough trees" | 5 | 4 | 3 | 2 | 1 | $TOSI = \dfrac{(Q1 + Q2 + Q3)}{3}$ |
| Q3: "The trees in my neighborhood are in good condition and healthy" | | | | | | |
| Q4: "The city should maintain all trees along the street" | | | | | | |
| Q5: "The city should prioritize maintenance of trees along the street in low-income communities" | 5 | 4 | 3 | 2 | 1 | $TMSI = \dfrac{(Q4 + Q5 + Q6)}{3}$ |
| Q6: "The city should plant trees in all available spaces along the street" | | | | | | |

One open-ended question asked the participants about municipality strategies to increase tree canopy [Appendix A, question 7]. After observing the responses, we coded repetitive values from the answers and created a typology based on UES and management challenges. Table 2 shows the coded values that translated the public concerns about ecosystem services, tree maintenance and financial concerns.

The last survey question about urban forestry asked if participants had trees on their property [Appendix A, question 8]. We measured the answers per zip code, informing the percentage of participants that had trees on their yards.

Four questions explored the socioeconomic characteristics of the participants [Appendix A, Questions 9–12]. We asked questions about the participants' race, income and zip code for the socioeconomic indicators. These indicators were traditional demographic parameters in other urban forestry and ecosystem services studies [8,10,12]. We also asked about housing ownership, as house tenure is an indicator of membership and active participation in urban environmental planning [41]. We compared the survey data's socioeconomic indicators to the census data to reinforce the survey sample's validity.

**Table 2.** Coded values from the open-ended question regarding strategies to increase tree canopy.

| Type | UES | Typology |
|---|---|---|
| Regulation and Provisioning services | Climate | Support microclimate, provide shade, mitigate urban heat island |
| | Air Quality | Promote clear air, mitigate pollution on transit corridors and industrial areas |
| | Water flow | Encourage tree planting in water facilities, acknowledge that trees consume water and support maintenance of groundwater |
| | Water purification | Improve watersheds and riparian areas, industries that pollute water should contribute to tree maintenance |
| | Erosion | Awareness that trees prevent landslides risks |
| | Natural Disaster Regulation | Flood prevention, stormwater mitigation in green infrastructures, reduction of impervious surfaces |
| | Pollination | Support bees and pollination |
| | Pest and disease | Physical and mental health, elders benefit from trees, industries that use pesticides should contribute to tree maintenance |
| | Waste | Pollution mitigation, energy savings, households with trees should have a discount on sewage bills, companies with more waste generation should contribute to tree maintenance |
| | Food | Encourage planting of fruit trees, urban agroforestry, mitigation of food deserts |
| Cultural services | Recreation and tourism | Recreation and relaxation, preference for trees over recreation facilities, canopy attracts tourists to the city |
| | Aesthetics and inspiration | Inspiration, beautification, common appreciation for trees |
| | Knowledge Systems | Educate people on how to plant and maintain trees, partnership with education institutions, job creation, encourage workshops about the importance of trees |
| | Religious and spiritual | Biophilia, spirituality, partnership with religious groups |
| | Cultural heritage | Cultural values of trees, community bonding, civic engagement, diversity of cultures and their interpretation of trees |
| | Natural heritage | Promote native trees, promote wildlife habitat, canopy preservation, encourage natural heritage stewardships |
| Management Challenges | Financial solutions | Tree giveaways, public/private/nonprofit partnerships, volunteers to reduce budget costs, use of taxes/donations/fundraisers resources |
| | Financial burden | Tree permit taxes, maintenance costs, other civic priorities over trees |
| | High maintenance | Right tree/right place, water and pruning care, public utilities, sidewalk maintenance, planning for climate change, regulations for trees in developed areas, tree maintenance strategies |

2.2.3. Survey Data for Public Perceptions of Urban Forestry

To integrate the analysis from the NLCD tree canopy, census and survey datasets, we developed a conceptual model that described the analytical steps for addressing our research aims (Figure 2). The conceptual model contained, at its core, the research question: *How do neighborhood trees and socioeconomic indicators mediate the public perceptions of available ecosystem services?* The conceptual model also integrated the datasets vis-a-vie specific questions that reference each of the three datasets we employed:

1. Census data and socioeconomic survey questions: Does the variation of socioeconomic indicators in the survey provide a good representation of the census data?
2. Census data and tree canopy data: Is there a relationship between socioeconomic indicators and tree canopy?
3. Tree canopy data and urban forestry survey questions: Does the presence of trees influence the public perceptions of urban ecosystem services?

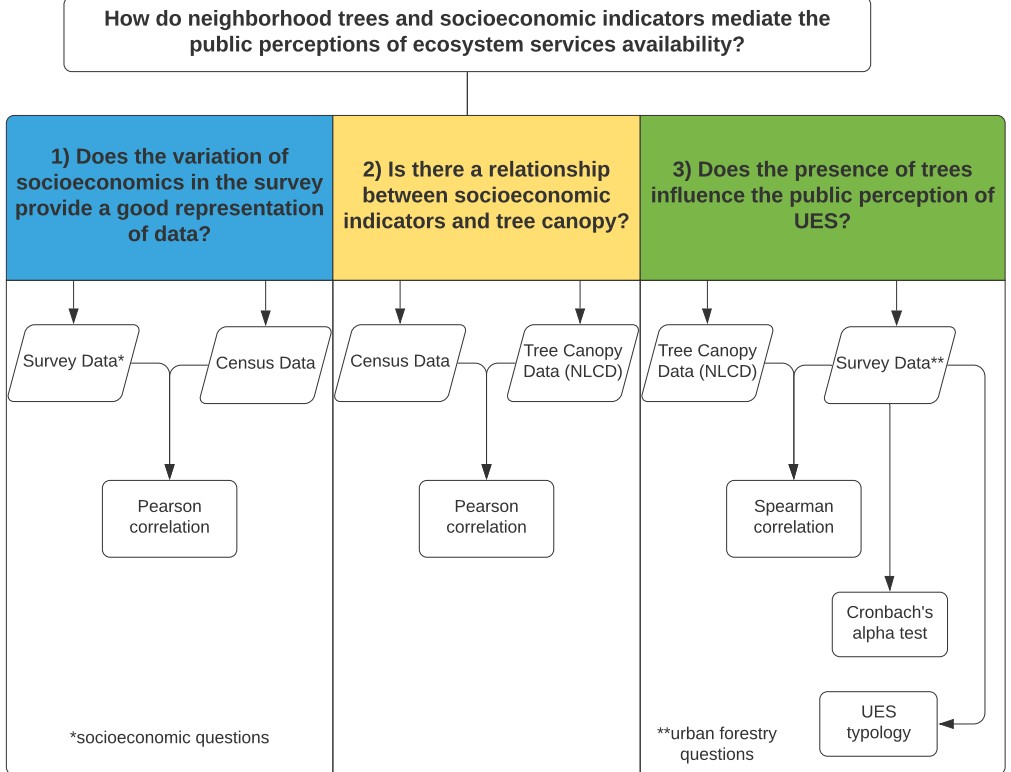

**Figure 2.** Flowchart with the research questions and research design. * survey data related to socioeconomic questions; ** survey data related to urban forestry questions.

To answer the first two questions, we performed the Pearson correlation test between the socioeconomic survey questions and Census data and between tree canopy data and Census data, as the variables were parametric. To answer the third question, we performed the Spearman correlation test between the tree canopy data and the survey urban forestry questions, as the variables were nonparametric. For both Pearson and Spearman correlation we considered the results that were significant with $p < 0.05$. For the urban forestry questions in the survey that used a Likert scale (Table 1), we performed Cronbach's alpha, which is a reliability test for the internal consistency of scaled questions and their variance. All statistical tests were performed with SPSS software v.26. Using the open-ended responses, we created a UES typology table (Table 2), which also served as additional data for evaluating and corroborating the statistical analysis.

### 3. Results

The survey for public perceptions of urban forestry and UES had 2548 valid answers from 26 zip codes within Portland. The survey responses ranged between 15 and 249 answers per zip code [Appendix B, Figure A1; Appendix B, Table A1]. As a voluntary online survey distributed in community engagement platforms (municipal listserv, Nextdoor, social media channels, focus groups, public meetings), there was a high chance that the participants had previous interests in urban forestry and city planning. We obtained completed answers and discarded those who did not complete the survey.

In the following subsections, we will answer the specific research questions regarding the correlation between the variables of census data, survey data and tree canopy data. In the last subsection, we will summarize the core question "*how do neighborhood trees and socioeconomic indicators mediate the public perceptions of ecosystem services availability*?" using the open-ended statements and their associations with the statistical findings.

### 3.1. Does the Variation of Socioeconomics in the Survey Provide a Good Representation of the Census Data?

To answer this question, we compared the census data and survey's socioeconomic indicators, which were both collected in 2017 (Table 3). The correlation between survey answers and the number of households per zip code was moderately strong (R = 0.554) and significant ($p < 0.01$). This result suggested that the number of respondents reflects the total population size within the zip codes.

We found strong Pearson correlation values between the census and the survey for the variables of house ownership (R = 0.796; $p < 0.01$) and income (R = 0.922; $p < 0.01$). The percentage of house ownership was higher among the survey respondents (82.09%) than the values indicated by the census data (51.54%). We believed that our survey targeted participants aware of the local public budget [39], as property owners have more responsibility with taxes that support tree maintenance.

For the race variable, both data from the census and the survey showed that Portland has a majority white population in all zip codes. Due to this fact, we labeled all non-white races as people of color. People of color (POC) is a term commonly used in the US to describe a population that is not white. The correlation values between the census and survey data for the POC variable was moderate (R = 0.402, $p < 0.05$). The percentage between POC in the census (22.22%) and survey (23.29%) had similar values.

Overall, the results suggest that for the specific characteristics the survey contained a representative sample for the city as a whole, which provides support to address the remaining questions. Our survey had a consistent representation with the Census data, with significant values for population size, race, income and house ownership.

**Table 3.** Pearson correlation values between socioeconomic indicators from the survey and census.

| Variable | Mean Value Per Zip Code | Pearson Correlation |
|---|---|---|
| Surveys (N) | 98 ± 66 | 0.554 ** |
| Households (N) | 11318 ± 5076 | |
| Housing ownership census (%) | 51.54 ± 15.20 | 0.796 ** |
| Housing ownership survey (%) | 82.09 ± 12.87 | |
| People of color census (%) | 22.22 ± 8.56 | 0.402 * |
| People of color survey (%) | 23.19 ± 7.26 | |
| Mean income census (US $) | 87,813.04 ± 28,397.40 | 0.922 ** |
| Mean income survey (US $) | 93,319.65 ± 20,275.61 | |

* Level of significance = 0.05; ** level of significance = 0.01.

### 3.2. Is There a Relationship between Tree Canopy and Socioeconomic Indicators?

Earlier research from several studies across the United States [6–8] and Portland [36] suggested that historically underserved communities are less likely to have immediate

access to tree canopy. In contrast, white and wealthier populations have greater access, partly due to historical policies that segregated neighborhoods [38]. This study was no exception and corroborated previous findings. Using Equation (1), we observed that zip codes in NW and SW had higher tree canopy than zip codes in the eastern sectors (E, N, NE, SE) of the Willamette River. NW and SW had respectively 42.5% and 37% of canopy cover and $125,739 and $100,696 of household incomes (Table 4). East had the lowest income ($57,104) and 12.4% of average canopy. The values of household income were obtained from the census data.

**Table 4.** Average tree canopy and income within the zip codes sectors in the study area.

| Variables | E | N | NE | NW | SE | SW |
|---|---|---|---|---|---|---|
| Canopy | 12.39% | 7.80% | 7.30% | 42.45% | 10.24% | 37.02% |
| Income | $57,104 | $86,824 | $95,714 | $125,739 | $93,200 | $100,696 |

Pearson correlation values between tree canopy and census socioeconomic indicators for income, race and house ownership (Table 5). The strongest correlation across all the sociodemographic was between tree canopy and income (R = 0.625, *p* < 0.01). This result supported the findings observed in Figure 1 and Table 4, with a high percentage of canopy cover in affluent neighborhoods of West Portland, suggesting that people with higher income in Portland have more access to the urban tree canopy. The results for the correlation between tree canopy with house ownership (R = 0.206; *p* > 0.3) and race (R = −0.186; *p* > 0.3) did not have significant values. As such, we conclude that income is the only significant (and positively correlated) variable in relation to the amount of tree canopy for the study area, as found in other urban forestry studies that used Portland as a case study [10,38,42].

**Table 5.** Pearson correlation values between tree canopy and census socioeconomic indicators.

| Correlation Variables | Pearson Correlation |
|---|---|
| Tree Canopy and Income | 0.625 ** |
| Tree Canopy and Race | −0.186 |
| Tree Canopy and House ownership | 0.206 |

** Level of significance = 0.01.

*3.3. Does the Presence of Trees Influence the Public Perception of UES?*

3.3.1. TOSI and TMSI Indicators

We combined three questions to build the TOSI, regarding: (1) the satisfaction with the number of trees; (2) the good condition of trees and (3) the importance of trees. The TMSI combined three questions about: (4) the maintenance of street trees; (5) the maintenance of trees in low-income communities and (6) the planting of new trees. Table 6 shows Cronbach's alpha results for the questions that encompassed the indexes.

While the Cronbach's alpha value of 0.490 can increase to 0.612 by removing Question 2 of the TOSI and TMSI multi-metric, we maintained the question because only 78.63% of respondents addressed all six questions, while 8.70% respondents answered five questions, 6.98% answered four questions, 5.61% answered three questions and 0.08% answered two questions. While multi-metric methods (ecosystem services coding, Likert scale, TOSI, TMSI) are reduced by including additional questions, some of which may not be addressed, doing so also increases the diversity and reliability of responses. In addition, surveys about public perceptions of urban forestry are relatively limited and the development of such metrics and observations, we expect, can contribute to further comparative studies.

**Table 6.** Cronbach's alpha for the Likert scale questions of survey.

| Total Statistics Cronbach's Alpha = 0.490 | Item Statistics | | |
|---|---|---|---|
| Questions | Mean | Std. Deviation | Cronbach's Alpha If Item Deleted |
| Q1: "Portland's trees are important to me" | 4.81 | 0.49 | 0.502 |
| Q2: "My neighborhood has enough trees" | 3.07 | 1.26 | 0.612 |
| Q3: "The trees in my neighborhood are in good condition and healthy" | 3.39 | 0.91 | 0.505 |
| Q4: "The city should maintain all trees along the street" | 3.70 | 1.23 | 0.287 |
| Q5: "The city should prioritize maintenance of trees along the street in low-income communities" | 3.98 | 1.2 | 0.287 |
| Q6: "The city should plant trees in all available spaces along the street" | 3.61 | 1.32 | 0.340 |

The variation of TOSI and TMSI answers are presented in a Likert scale across a series of maps (Figure 3). The Likert scale ranged from 1 to 5, where 1 represents strongly disagree and 5 represented strongly agree. The TOSI and TMSI were the three questions' [Appendix A, Questions 1–6] average values combined on the respective indexes.

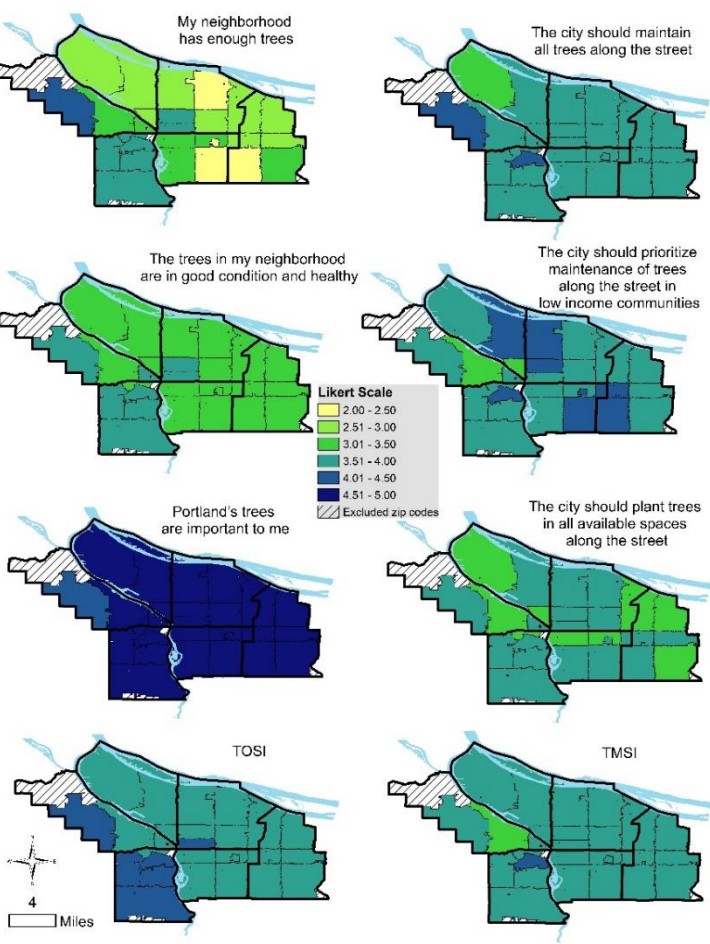

**Figure 3.** Range of answers to individual Likert scale questions that are used to generate the multimeric indices of ownership [TOSI] and maintenance [TMSI] for the study area.

In the questions associated with TOSI and TMSI, most of the answers per zip code were higher than 3 on the Likert scale, except for satisfaction with neighborhood trees. The eastern zip codes had lower satisfaction with the number of trees than the western zip codes. In the question about tree health, the western zip codes had a higher rate on the

Likert scale. A zip code in NW, a high-income and high canopy part of the study area, scored the lowest value for trees' importance.

### 3.3.2. Public Perceptions of Urban Ecosystem Services

The results indicated that most of the participants had trees on their private property. The average percentage of participants that informed having trees on their yard was 94.2%. The lowest rate of private property trees was 75.8%, in the zip code 97209, an urban renewal area (redevelopment of industrial and low-income areas in inner-city) in Northwest Portland [31].

The last question about urban forestry perceptions was open-ended and we coded the answers using UES typology (Table 7). According to the coded answers, the strategies recommended a focus on urban forestry management (56.1%), cultural ecosystem services (31.5%) and regulating and provision services (12.4%).

**Table 7.** Public perceptions of relevant strategies to increase urban tree canopy.

| Type | UES | Survey Answers (%) | Survey Answers Aggregated (%) |
|---|---|---|---|
| Regulating and provisioning ecosystem services | Climate | 1.67 | 12.36 |
| | Air Quality | 2.67 | |
| | Water flow | 2.02 | |
| | Water purification | 0.67 | |
| | Erosion | 0.19 | |
| | Natural hazard | 1.14 | |
| | Pollination | 0.16 | |
| | Pest and disease | 2.00 | |
| | Waste | 1.02 | |
| | Food | 0.81 | |
| Cultural ecosystem services | Recreation and tourism | 0.70 | 31.54 |
| | Aesthetics and Inspiration | 3.18 | |
| | Knowledge | 11.01 | |
| | Religious and spiritual | 0.33 | |
| | Cultural heritage | 9.55 | |
| | Natural heritage | 6.78 | |
| Management Challenges | Financial Solution | 33.59 | 56.10 |
| | Financial Burden | 4.53 | |
| | High Maintenance | 17.98 | |

Within the management challenges, 33.6% of the answers suggested financial solutions. The answers recommended the municipality to seek partnerships with volunteer associations, donation of seedlings and financial support for homeowners and renters, such as tax and water/sewage bill discounts. The second most mentioned typology was the maintenance of trees, with 18% of answers showing that respondents were aware of tree health and upkeep's essential requirements. The participants informed that proper tree pruning, tree species selection, tree debris removal and regulation for trees in developed areas were their top priorities.

Among the UES, knowledge systems and cultural heritage were the most significant concerns, holding 11% and 9.6% of the answers, respectively. Knowledge systems associated with urban forestry solutions that educate the population on how to take care of trees. Connection with teaching opportunities through schools and training programs can

prepare present and future generations to maintain trees and understand urban ecology interactions. Awareness of cultural heritage indicated the respondents' related urban forestry to community development, community engagement and neighborhood pride. The open-ended answers recommended tree planting events to promote activities that bring interaction among neighbors to praise trees' values for multiple cultures and ethnicities.

As most of the answers indicated strategies using financial solutions and tree maintenance, we conducted a separate analysis only with the UES typology. Excluding the management challenges (Table 7), knowledge systems and cultural heritages lead the answers with 25.1% and 21.8% of the responses, respectively. Natural heritage was the third most important, with 15.5% of answers. Natural heritage responses concerned the loss of mature trees, biodiversity and wildlife habitat. The respondents commented that small trees could take longer to provide the ecosystem services promoted by centenary trees susceptible to removal for new developments or infrastructure challenges.

The final analysis described the Spearman correlation between survey answers and the percentage of tree canopy per zip code (Table 8). The variables that had the strongest positive correlation with tree canopy were satisfaction with neighborhood trees (R = 0.767), satisfaction with tree health (R = 0.704), TOSI (R = 0.758) in a 0.01 significance level and aesthetics and inspiration (R = 0.453) in a 0.05 significance level. Though these are general findings, we note that these levels of significance and strength of the relationship varied by zip code.

**Table 8.** Correlation between tree canopy and survey answers about public perceptions of urban forestry.

| Public Perceptions | Spearman Correlation (R) | Public Perceptions | Spearman Correlation (R) |
|---|---|---|---|
| Climate | −0.136 | Natural heritage | −0.191 |
| Air quality | 0.226 | Financial Solution | 0.039 |
| Water flow | 0.009 | Financial Burden | −0.235 |
| Water purification | 0.194 | High Maintenance | −0.162 |
| Erosion | 0.289 | Food | −0.231 |
| Natural hazard | 0.034 | Trees on property | 0.048 |
| Pollination | −0.233 | Neighborhood trees | 0.767 ** |
| Pest and disease | 0.078 | Tree health | 0.704 ** |
| Waste | 0.158 | Importance of trees | −0.289 |
| Recreation and tourism | −0.037 | TOSI | 0.758 ** |
| Aesthetics and inspiration | 0.453 * | Street trees | 0.241 |
| Knowledge | 0.227 | Trees in low income areas | −0.27 |
| Religious and spiritual values | −0.149 | Plant street trees | −0.104 |
| Cultural heritage | −0.009 | TMSI | 0.005 |

* Level of significance = 0.05; ** Level of significance = 0.01.

### 3.4. How Do Neighborhood Trees and Socioeconomic Indicators Mediate the Public Perceptions of Ecosystem Services Availability?

Seven out of eight zip codes in Portland's western sectors had a canopy rate higher than the eastern sectors. The justification for the uneven distribution is associated with the early conservation practices of urban forestry in the western zip code sectors [23], the hilly geological formation [26,27] and the higher income [8]. Tree canopy had a significant positive relationship to TOSI indicators and cultural UES of aesthetics and inspiration (Table 8). However, the excessive canopy does not please everyone, as expressed in the following statements from participants living in high canopy areas:

Too many trees already. While they have benefits, the trees need to be healthy and co-exist safely with residents. This requires regular, vigilant maintenance,

which a lot of people (...) fail to do. We've repeatedly witnessed the tragedy of human deaths and property destructions, especially this last winter. Even one death is too many! We need to take better care of the existing trees before we consider adding more. (97221—Southwest)

I don't necessarily think the city should plant more trees. While the trees here certainly help relieve heat we need to be mindful of how little sun we get here (in Portland rainy weather). I think the city needs to maintain a balance between densely wooded areas (e.g., Forest Park) and highly exposed areas (e.g., central eastside). I think students and volunteers could plant lots of trees. (97221—Southwest)

(...) determine the most aesthetic and functional places to plant trees and then only plant trees that make logical sense for the conditions present in the chosen locations. (97210—Northwest)

I think the city should plant fruit and nut trees when they plant trees. They grow just as easy as any other tree. Most have beautiful flowers and foliage. And better yet they make healthy snacks especially in low income neighborhoods and food deserts. (97236—East)

(...) I can't see cars coming at intersections because there are too many trees already in my neighborhood. There is a near miss almost every day at my house because people can't see the cars coming. (97212—Northeast)

East of the Willamette River, five zip codes stood out with more than 10% of canopy cover. The zip code 97236 had 23.5% canopy cover in the Pleasant Valley area, an early incorporated neighborhood near an affluent suburb, Happy Valley. This zip code area also bears Powell Butte Natural Area, a remaining forest fragment. 97212 had 17% of tree coverage and the fourth largest average income citywide. 97202 had 13.8% of canopy cover and a protected riparian zone in the eastbound of Willamette River. 97215 had 13% of canopy cover and a preserved forest fragment on Mount Tabor Park. 97266 had 12.4% of tree coverage and was the third-lowest income zip code. However, it bears a forest fragment on Kelly Butte Natural Area. These findings showed that tree canopy follows income and geological formation features, such as riverbanks, forest fragments and hilly areas.

In low-income communities, trees' maintenance is observed as a financial burden, which can be classified as an ecosystem disservice [43]. Two zip codes from the East sector answered that 8.3% and 7.7% of the increasing tree canopy strategies have financial burden as a management challenge. The average answer mentions for financial burden was 4.5% per zip code. Seven out of eleven neighborhoods with answers above average are in the East, the sector with the lowest income and low canopy (Table 4). The following statements extracted from the open-ended question about strategies for increase tree canopy reflect the concerns for financial burden within residents of low-income neighborhoods:

"Offer to plant them (trees), offer low income solutions to families" (97233—East)

(...) lots of trees (are) in bad places and they die so better planning would do just fine and offering classes for ppl (people) who want to learn how to maintain the trees better and if they have it why does low income not have access to the classes and knowledge of them? (97233—East)

Don't charge for leaf cleanup. Offer a small tax credit for trees planted and maintained to property owner(s). Education regarding the importance of trees for everyone. Offer education to grow trees in a pot. Then everyone can grow a tree. (97220—East)

(...) more financial and volunteer support to groups like that (street tree planting nonprofit). Also when Portland had the ice storm this past winter, many residential trees came down. (...) people could bring downed trees and branches, maybe

even for a donation. Free mulch for the city and donations! Tree culture need(s) to be supported in more ways than just plantings (...) to make owning trees easy. (97220—East)

I'm sure a lot of people are scared away from that program (street tree planting program) due to the need to care for the tree and the possible damage to sidewalks that they will eventually be forced to repair at their expense later on down the line. (97220—East)

The answers for increasing canopy strategies also indicated concerns for other UES, such as climate change, knowledge systems, natural heritage and cultural heritage. Excluding the answers about management challenges, responses about regulating and provisioning UES represented 28% and cultural UES accounted for 72% of answers. We believed that people highly value cultural benefits from trees in urban areas due to the "extinction of experience" [3]. Within cultural ecosystem services, we distinguished patterns in responses that seek environmental education, multiple ethnic values for forest biodiversity and conservation of heritage trees. The answers associated with knowledge systems indicated that besides incentives for tree planting, people also need to know how to care for trees and their importance regarding ecosystem services. Public surveys assessing urban forestry and management of UES have suggested the enforcement of knowledge systems [44]. As observed in the previous statements, the survey participants repetitively suggested partnerships with education institutions, urban forestry jobs, internships and free workshops. The responses indicated that the residents expect more personal accountability for ownership if they have access to knowledge, tools and technical support from municipality and nonprofits.

In answers that mentioned cultural heritage, people requested more planting events to bond with neighbors and create a sense of community. The participants asked for multilingual tree support, public participation in urban forestry planning and access to trees with ethnic values regarding inclusion and diversity measures. Natural heritage answers indicate the population's willingness to perpetuate biodiversity, urban ecology and centenary trees. Together, cultural and natural heritage are UES that reflect landscape interpretations, which are individual perspectives of the environment based on personal background, memories, experiences and expectations. In general, environmental planning bears management tools that can perpetuate systemic racism by restricting access to ecosystem services based on socioeconomic values, reducing maintenance costs in areas with low-income and people of color and not acknowledging the diversity of behaviors in public space [45,46]. Surveys, interviews and focus groups can collect ideas, perspectives and expectations from historically unheard voices and open a path for public participation in urban forestry planning.

Portland had a complex history of gentrification that burdened the black community with displacement, loss of sense of spatial identity and identity representation [21]. The increase in population promoted real estate development for housing, business and other civic infrastructures. In Portland, there is an inverse relationship between canopy cover and urban development indicators, as water pipers [10]. The survey answers indicated that people are aware that new developments threaten trees, impacting their maintenance and natural heritage. The following statements are from the zip codes with lower housing ownership [Appendix B, Table A1] and most gentrified areas [32]:

"Yes, we need many more trees but (...) focus on protecting the most mature trees as they have been shown to provide the greatest benefits." (97227—North)

Demolition review to ensure maintenance of entire tree canopy as development can remove existing trees. The accelerated development in Portland has not been counterbalanced with a comprehensive plan to prevent tree removal and plant new trees. It has greatly reduced potential green spaces which could offset somewhat the unbridled concrete development. (97232—Northeast)

Trees are natural green infrastructures that support stormwater catchment, avoid erosion and improve air quality. Other forms of green infrastructure such as rain gardens, green roofs, artificial wetlands and parks can bring green gentrification—a gentrification process caused by the implementation of green infrastructures. The following Discussion section will explore how the results are associated with environmental justice and landscape management.

## 4. Discussion and Conclusions

This study aimed to address questions about the relationship between the existing amounts of neighborhood tree canopy with sociodemographic data and community perspectives. One of the explicit goals of the present study was to understand the relationship between tree ownership, maintenance and the amount of tree canopy. We found that zip codes with higher tree canopy were consistent with greater sense of ownership and quality of trees, as measured by the TOSI. Specifically, the two TOSI questions about the number of trees and the good condition of trees had a strong correlation and high significance values with tree canopy. This finding is consistent with a low Cronbach's alpha, suggesting that this metric can be explored further, perhaps in a different setting.

Our findings also indicate that a sense of ownership comprises the importance and satisfaction with the quantity and the quality of trees in the neighborhood, including trees in private property, public spaces and the right-of-way. Affluent zip codes had higher canopy cover had a higher correlation with public maintenance of street trees, as measured by the TMSI. While earlier research suggests that tree canopy and income are correlated in the U.S. [8–10], the perception of tree ownership is a new concept that brings accountability of ownership and maintenance in relation to urban ecosystem services. We observed a lower correlation between the tree canopy and TMSI than with TOSI (Table 8). TMSI also presented a lower range of mean values on the Likert scale response (Table 6), suggesting a common concern about tree care citywide (Figure 3). In the question about increasing tree canopy strategies, the responses about tree maintenance represented about 18% of the answers (Table 7).

Perhaps one of the most germane findings in our study is the fact that while a canon of literature describes the importance of trees in providing UES (e.g., pollination, air quality, climate regulation, etc.), our survey findings indicate that issues about management and cultural ecosystems services feature prominently among the respondents. This finding, while perhaps mundane, is significant for several reasons, including the fact that respondents seem to recognize the financial burden and maintenance when considering trees. If this finding is consistent across the city, then municipal goals for expanding tree canopy will face formidable obstacles if they present trees an important for traditional regulating ecosystem services. Rather, recognizing that communities are considering, perhaps less these regulatory services, than those surrounding maintenance and financing, may provide more effective.

Suitable messaging may not be the only implication of this finding. If aesthetics, inspiration and level of ownership are correlated with the amount of neighborhood tree canopy (Table 8), then attempts to create distributional equity will require considerable recognition of the maintenance costs involved. Maintenance often includes the planting appropriate species, pruning of trees, watering and a host of other monitoring to ensure healthy growth. Responses indicated the importance of maintenance and also suggested that municipalities provide support to those communities how may not have the financial resources to address maintenance concerns. Currently the City of Portland requires adjacent property owners to maintain all public trees, which increases the level of inequity already experienced by lower income communities. Perhaps the positive and significant correlation between income and presence of tree canopy is because lower income community may not prefer trees due to the cost of maintenance, which the open-ended responses indicated. The development of trees in the right-of-way interacts with other infrastructure, such as sidewalks, residences and transit features. Studies that observe the growth and health of

street trees [47], suggest regular monitoring and maintenance such as root pruning and interaction with underground infrastructures [48–50] which can help to ensure a healthy and robust tree canopy.

While these findings offer a first step towards integrating community perspectives into urban forest management, the findings suggest the importance of engaging communities in the management of tree canopy. The open-ended results suggested that respondents genuinely understand the challenges facing urban forest management and the importance of finding systematic ways to maintain canopy and provide equitable access to all residents. Our findings indicate, for example, that promoting financial solutions that optimize public and private budgets toward urban forestry and cultural heritage practices that engage the community in participatory planning and empowerment are the priority strategies for increasing tree canopy. With the multiple goals for achieving environmental equity through urban forestry, these strategies must also include raising awareness about the inequitable distribution of existing tree canopy, planting more trees in vulnerable communities, exploring diverse perspectives about climate resilience and exploring the role of trees among historically marginalized communities [46].

This study offers a means for understanding the importance of ownership and maintenance in addressing urban ecosystem services. While our survey can help to underscore some of these priorities, we recognize that engaging communities about urban forestry may pose several challenges. If employed effectively, other data collection methods, including listening sessions, focus groups and interviews, can help to contextualize urban forestry within the broader set of community needs that may be priorities. The COVID-19 pandemic has made clear that priorities such has housing, food and medical care are often front-and-center among POC and lower income communities, which may pose several challenges for discussions about urban forestry. Since POC and low-income neighborhoods have been excluded from planning for a healthy and abundant urban forest [8], a pattern that may be associated to redlining practices in the U.S. [51] perhaps the built and natural environment in neighborhoods can be a direct means for understanding other pressing priorities. By engaging historically disinvested communities and address distributional injustices that have created current inequities in the distribution of tree canopy cover, municipal agencies may find creative solutions that 'multi-solve' the myriad pressing challenges.

Our study found that survey respondents seeks more measures and strategies to address cultural UES, such as cultural heritage, natural heritage, aesthetics and inspiration. The gap of systematic descriptions for these cultural UES within municipal plans may require greater levels of public involvement, which would build on diverse perspectives [46]. While government agencies are often responsible for the management of public spaces, the same agencies may not be trusted allies with communities that have been historically disinvested. As such, management options that engage community-based organization (CBOs) may be a more trusted and effective approach for soliciting plausible solutions. Such CBOs can work with community members to explore their expectations for land use, tree canopy, species selection, planting events, tree giveaways and volunteer workforce.

Examples of such cross-sectoral urban forestry management are emerging. In Toronto, CBOs had a more diverse species list than municipal agencies, landscape architects and nurseries [52]. In Detroit, interviews with CBO staff and recipients of giveaway trees informed that the ability to choose the tree species is a fact that impacts the willingness of residents to care for private trees, as well as live in areas with lower canopy. In both cases, studies have found that the major challenges are concerns with maintenance practices and costs, such as pruning, sidewalk damages and tree debris removal [53]. Both studies suggested stewardships for a functioning and healthy urban forestry, where CBOs would have the goal to promote understanding, while supporting cultural ecosystem services.

**Author Contributions:** L.A.C.N. have performed the analysis and wrote the manuscript. V.S. have collected the survey data, coordinated the researched, and reviewed the manuscript. Both authors have read and agreed to the published version of the manuscript.

**Funding:** This research was funded by Conselho Nacional de Desenvolvimento Científico e Tecnológico (CNPq), grant number: 221077/2014-6—LASPAU—Brasil/CNPq—GDE—EUA; and by the US Forest Service's National Urban and Community Forestry Challenge Grant (17-DG-11132544-014) and the Robert Wood Johnson Climate and Health project.

**Conflicts of Interest:** The authors declare no conflict of interest. The founding sponsors had no role in the design of the study; in the collection, analyses, or interpretation of data; in the writing of the manuscript, and in the decision to publish the results.

## Appendix A

**Survey:**

Tree Ownership Satisfaction Index (TOSI):

Q1 Portland's trees are important to me. Strongly Agree/Agree/Don't Know/Disagree/Strongly Disagree.

Q2 My neighborhood has enough trees: Strongly Agree/Agree/Don't Know/Disagree/Strongly Disagree.

Q3 The trees in my neighborhood are in good condition and healthy. Strongly Agree/Agree/Don't Know/Disagree/Strongly Disagree.

Tree maintenance Satisfaction Index (TMSI).

Q4 The city should maintain all trees along the street (in the public right-of-way, next to the sidewalk area): Strongly Agree/Agree/Don't Know/Disagree/Strongly Disagree.

Q5 The city should prioritize maintenance of trees along the street (in the public right-of-way, next to the sidewalk area) in low-income communities: Strongly Agree/Agree/Don't Know/Disagree/Strongly Disagree.

Q6 The city should plant trees in all available spaces along the street (in the public right-of-way, next to the sidewalk area): Strongly Agree/Agree/Don't Know/Disagree/Strongly Disagree.

Strategies for increase tree canopy:

Q7 How do you think the city should get more trees planted?

Presence of trees in private properties:

Q8 Do you have trees at the property where you live? Yes/No.

Demographic Questions:

Q9 What is your household income? Less than $10,000/$10,000–$19,999/$20,000–$29,999/$30,000–$39,999/$40,000–$49,999/$50,000–$59,999/$60,000–$69,999/$70,000–$79,999/$80,000–$89,999/$90,000–$99,999/$100,000–$149,999/$150,000–$199,999/$200,000 or more/I don't know.

Q10 Which best describes your race or ethnicity? Choose as many as apply:

❑ Alaska Native ❑ American Indian/Native American ❑ East Asian ❑ South Asian ❑ Southeast Asian ❑ West Asian ❑ Middle Eastern ❑ Black or African American ❑ African ❑ Hispanic or Latino ❑ Native Hawaiian or Pacific Islander ❑ Slavic or Eastern European ❑ White ❑ Other (please specify).

Q11 What is your home zip code?

Q12 Do you rent or own the place where you live? Rent/Own.

## Appendix B

**Table A1.** Socio indicator data from survey and Census.

| Zip Code | Area | Valid Survey Answers (N) | Population Census (N) | Households Census (N) | Housing Ownership Census (%) | Housing Ownership Survey (%) | Housing Ownership Census (N) | Housing Ownership Survey (N) | People of Color Census (%) | People of Color Survey (%) | People of Color Census (N) | People of Color Survey (N) | Mean Income Census (US$) | Mean Income Survey (US$) |
|---|---|---|---|---|---|---|---|---|---|---|---|---|---|---|
| 97201 | SW | 50 | 17,566 | 9009 | 31.58 | T52.00 | 2845 | 26 | 23.17 | 26.53 | 4070 | 13 | 92,276 | 88,666 |
| 97202 | SE | 211 | 42,189 | 18,135 | 49.68 | 81.52 | 9010 | 172 | 14.47 | 13.74 | 6103 | 29 | 92,391 | 96,865 |
| 97203 | North | 107 | 34,089 | 12,091 | 55.43 | 85.98 | 6702 | 92 | 24.2 | 21.5 | 8250 | 23 | 67,963 | 80,663 |
| 97205 | SW | 31 | 7122 | 4881 | 17.62 | 58.06 | 860 | 18 | 20.33 | 10 | 1448 | 3 | 61,537 | 81,087 |
| 97206 | SE | 249 | 50,655 | 20,100 | 62.83 | 84.34 | 12,628 | 210 | 19.94 | 34.94 | 10,102 | 87 | 71,658 | 83,602 |
| 97209 | NW | 33 | 16,507 | 11,376 | 25.58 | 60.61 | 2910 | 20 | 15.68 | 15.15 | 2589 | 5 | 82,582 | 100,645 |
| 97210 | NW | 35 | 11,676 | 6253 | 40.76 | 74.29 | 2549 | 26 | 9.76 | 25.71 | 1140 | 9 | 140,398 | 132,407 |
| 97211 | NE | 181 | 34,856 | 13,081 | 65.81 | 85.08 | 8609 | 154 | 27.21 | 19.89 | 9484 | 36 | 91,179 | 98,866 |
| 97212 | NE | 143 | 26,601 | 10,890 | 64.05 | 89.51 | 6975 | 128 | 15.81 | 11.89 | 4206 | 17 | 125,138 | 112,073 |
| 97213 | NE | 165 | 32,284 | 13,783 | 61.63 | 85.45 | 8495 | 141 | 15.98 | 17.58 | 5160 | 29 | 87,715 | 89,455 |
| 97214 | SE | 144 | 25,398 | 12,190 | 35.64 | 81.25 | 4344 | 117 | 12.91 | 22.22 | 3280 | 32 | 84,012 | 94,801 |
| 97215 | SE | 91 | 17,939 | 7802 | 63.86 | 92.31 | 4982 | 84 | 10.84 | 25.27 | 1945 | 23 | 101,953 | 97,530 |
| 97216 | East | 51 | 17,112 | 6530 | 46.66 | 90.2 | 3047 | 46 | 29.93 | 33.33 | 5122 | 17 | 53,870 | 70,306 |
| 97217 | North | 170 | 34,327 | 14,520 | 62.41 | 82.94 | 9062 | 141 | 22.34 | 25.29 | 7669 | 43 | 84,004 | 88,975 |
| 97218 | NE | 76 | 15,556 | 5543 | 59.1 | 90.79 | 3276 | 69 | 28.43 | 27.63 | 4422 | 21 | 68,539 | 73,450 |
| 97219 | SW | 202 | 41,534 | 16,561 | 70.32 | 90.1 | 11,646 | 182 | 13.05 | 17.82 | 5419 | 36 | 119,199 | 104,608 |
| 97220 | East | 93 | 30,374 | 11,488 | 56.27 | 91.4 | 6464 | 85 | 34.62 | 24.73 | 10,514 | 23 | 60513 | 80,632 |
| 97221 | SW | 41 | 12,363 | 5170 | 75.92 | 95.12 | 3925 | 39 | 11.79 | 34.15 | 1457 | 14 | 146,948 | 121,333 |
| 97227 | North | 15 | 4648 | 2259 | 31.03 | 86.67 | 701 | 13 | 24.94 | 26.67 | 1159 | 4 | 77,831 | 90,833 |
| 97229 | NW | 21 | 65,285 | 23,913 | 70.72 | 100 | 16,912 | 21 | 31.83 | 23.81 | 20,779 | 5 | 137,990 | 144,166 |
| 97230 | East | 77 | 39,884 | 14,967 | 56.97 | 93.51 | 8527 | 72 | 32.96 | 23.38 | 13,147 | 18 | 63,112 | 87,847 |
| 97232 | NE | 57 | 12,865 | 6641 | 31.86 | 56.14 | 2116 | 32 | 14.42 | 15.79 | 1855 | 9 | 86,325 | 104,727 |
| 97233 | East | 54 | 42,001 | 13,496 | 45.82 | 66.67 | 6184 | 36 | 35.16 | 37.04 | 14,767 | 20 | 49,769 | 51,875 |
| 97236 | East | 53 | 40,274 | 12,996 | 54.59 | 90.57 | 7094 | 48 | 36.56 | 30.19 | 14,726 | 16 | 60,022 | 69,200 |
| 97239 | SW | 67 | 16,682 | 8322 | 49.54 | 85.07 | 4123 | 57 | 17.89 | 14.93 | 2985 | 10 | 120,876 | 110,350 |
| 97266 | East | 131 | 34,757 | 12,280 | 54.24 | 84.73 | 6661 | 111 | 33.46 | 23.66 | 11,628 | 31 | 55,339 | 71,349 |

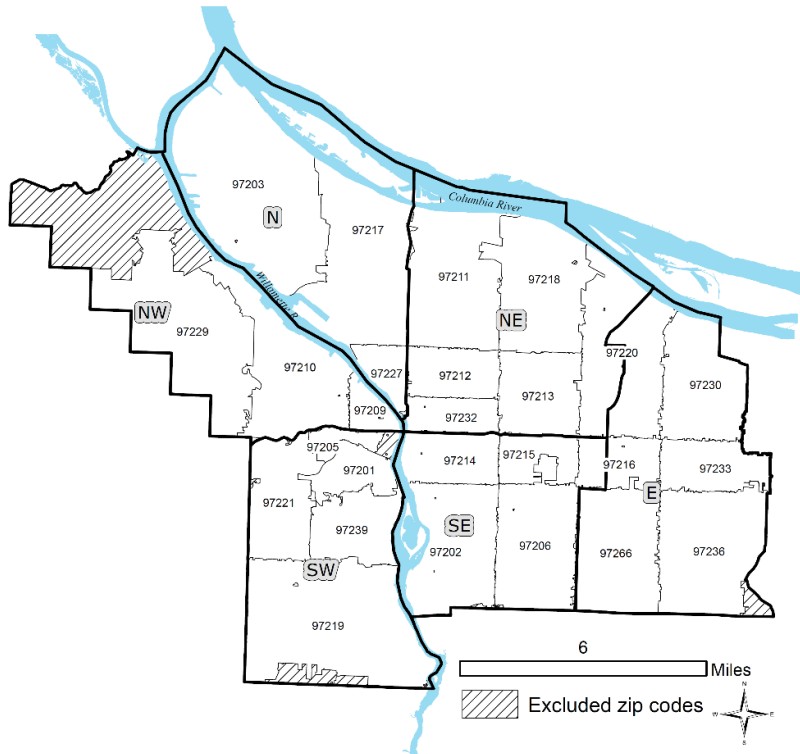

**Figure A1.** Zip codes in the study area.

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
