# Peer review of "Integrating Diverse Perspectives for Managing Neighborhood Trees and Urban Ecosystem Services in Portland, OR (US)"

_land, doi:10.3390/land10010048_

Round 1

Reviewer 1 Report

In Materials and Methods:

Due to the qualitative nature of some variables, it would be necessary to use Spearman's correlation coefficient or Kendall's tau. It is incorrect to use Pearson's correlation coefficient.

It should be explained in this section how the TOSI and TMSI indices are calculated, which are mentioned in the results.

It is necessary to give some measure of reliability of the survey, such as Cronbach's alpha, widely used in research studies with surveys and easy to interpret.

The Discussion is excessively long, as it is mixed with survey results. The discussion should be clearer.

Author Response

1. Due to the qualitative nature of some variables, it would be necessary to use Spearman's correlation coefficient or Kendall's tau. It is incorrect to use Pearson's correlation coefficient.

We are grateful to the reviewer for this suggestion, and have now included a Pearson correlation test between the survey socioeconomic questions and Census data, and between tree canopy data and Census data, as all these variables were parametric. We performed the Spearman correlation test between the tree canopy data and the survey urban forestry questions, as the variables were nonparametric.

2. It should be explained in this section how the TOSI and TMSI indices are calculated, which are mentioned in the results.

We have included additional and rephrased language to explain how the TOSI and TMSI indices were calculated. We describe that instead of analyzing the six questions separately, we created two multi-metric indexes: Tree Ownership Satisfaction Index (TOSI) and Tree Maintenance Satisfaction Index (TMSI). TOSI combined the three questions about tree ownership and TMSI the three questions about tree maintenance. TOSI and TMSI also used a Likert scale, and we calculated them by finding the average values of the three questions of each index, as displayed in Table 1. TOSI and TMSI resemble multi-metric indexes used in the biological assessment of watersheds. Such multi-metrics are common in several fields of study, and since each question offers a distinct perspective on ‘ownership’ and ‘maintenance,’ these indicienes provided a means to assess over-arching descriptions. 

We further specify that the first six questions of our study encompassed the perceptions of tree ownership and maintenance [Appendix A – questions 1 – 6]. Of those six, three questions are directly about tree ownership, including 1) the importance of trees, 2) the satisfaction with the number of trees, and 3) the satisfaction with trees' health [Appendix A, questions 1 – 3]. The other three questions are about tree maintenance, and include: 1) the level of maintenance of existing trees in the right-of-way, 2) the importance of maintaining trees in the right-of-way in low-income communities, and 3) the importance of planting new trees in the right-of-way [Appendix A, questions 4 – 6]. The participants used a Likert scale ranging between 1 and 5 to inform how much they agree or disagree with the six sentences related to tree ownership and tree maintenance. In the revised draft, Table 1 describes the range of Likert scale values, tree ownership and maintenance questions, and multi-metric evaluation. 

3. It is necessary to give some measure of reliability of the survey, such as Cronbach's alpha, widely used in research studies with surveys and easy to interpret.

For the urban forestry questions in the survey that used a Likert scale, we performed Cronbach’s alpha test, as the questions contributed for the two indexes TOSI and TMSI. We created an UES typology table based on the open-ended responses and we observed the answers’ statements to evaluate and corroborate the statistical analysis results. 

4. The Discussion is excessively long, as it is mixed with survey results. The discussion should be clearer.

We rearranged the results and discussion to better clarification. We added the open-ended quotes on results, and used the discussion section to compare studies and the impact of results. Now the results section is expanded and the discussion section is shorter. 

Reviewer 2 Report

The study provided a unique examination of spatial and psychological relationships between forests and people in urban setting. Overall, the study was well done and the paper well written, I have only a few minor comments. How was the survey advertised or pushed out to people? Were incentives used? This may help with getting better representation of non-white respondents. Was the survey pre-tested? How where the TOSI and TMSI indexes created? e.g., How do you know that importance, number of trees and condition are the only parameters important to satisfaction? Are the items based on related research or preliminary work (interviews?). Was the index validated using methods to determine internal consistency, such as Cronbach's alpha? Figure 2 seems to be missing A, B, C letters and there is a typo in the figure. This would be a good paper to cite in the discussion even though it was conducted in another city. Carmichael, C. E., & McDonough, M. H. (2019). Community stories: Explaining resistance to street tree-planting programs in Detroit, Michigan, USA. Society & Natural Resources, 32(5), 588-605. I did not see a conclusions section. This section and policy implications would be important for this paper.

Author Response

The survey was advertised through three specific sources: (1) online list-servs that reach anybody with an email and are part of the City of Portland’s public notices; (2) through the ‘Next Door’ list-serv that reaches approximately 20% of Portland residents (~100K); and (3) through social media. The survey was open from mid‐May to mid‐July 2017. The survey was also mentioned at the community advisory committee and focus group meetings. No incentives were used. 

The first six questions of our study encompassed the perceptions of tree ownership and maintenance Three questions about tree ownership asked the participants about 1) the importance of trees, 2) the satisfaction with the number of trees, and 3) the satisfaction with trees' health. Three questions about tree maintenance inquired about 1) the maintenance of existing trees in the right-of-way, 2) maintenance of trees in the right-of-way in low-income communities, and 3) planting new trees in the right-of-way. The participants used a Likert scale ranging between 1 and 5 to inform how much they agree or disagree with the six sentences related to tree ownership and tree maintenance. Instead of analyzing the six questions separately, we created two multi-metric indexes: Tree Ownership Satisfaction Index (TOSI) and Tree Maintenance Satisfaction Index (TMSI). TOSI combined the three questions about tree ownership and TMSI the three questions about tree maintenance. TOSI and TMSI also used a Likert scale, and we calculated them by finding the average values of the three questions of each index. TOSI and TMSI resemble multi-metric indexes used in the biological assessment of watersheds. 

For the urban forestry questions in the survey that used a Likert scale, we performed Cronbach’s alpha test, as the questions contributed for the two indexes TOSI and TMSI.  We created an UES typology table based on the open-ended responses and we observed the answers’ statements to evaluate and corroborate the statistical analysis results. 

We are grateful to the Reviewer for the suggestion of Detroit's paper, which we’ve now incorporated into our discussion session.

Reviewer 3 Report

The paper addresses an interesting topic, still there are issues that need to be strengthened or corrected, namely:

Literature review misses several seminal works and important advances crossing climate change and landscape planning.

Material and methods are hard to understand, because the research steps are not adequately described. I would recommend the introduction of a phased methodological diagram. The flowchart is somehow fuzzy.

The results are also not adequately presented. Figure 3 though interesting is much harder to read than a simple table. I believe further information is needed and climate change impacts need to be supported on further data.

Conclusions should be written on a more scientific way. As they are they highlight the limitations of the research.

Author Response

1. Literature review misses several seminal works and important advances crossing climate change and landscape planning.

This paper aimed to address the current needs of forest management by specifically integrating public perceptions into understanding urban ecosystem services (journal theme issue). While urban forest management is necessary to mitigate climate change and support climate resilience our aim is not directly address climate change, rather to examine whether trees are generally seen as favorable and relevant to greening efforts. Additionally, we decided to highlight how urban forest management addresses environmental justice and accessibility to ecosystem services in municipalities rather than focus majorly on landscape planning, which, as the Reviewer likely knows, is an entire field in and of itself. Using surveys and metrics to discuss the accessibility to ecosystem services and urban forestry implies a concern for climate change mitigation, though in the present paper does not center it. 

2. Material and methods are hard to understand, because the research steps are not adequately described. I would recommend the introduction of a phased methodological diagram. The flowchart is somehow fuzzy.

We expanded the methods and materials sections to include a flowchart for a better comprehension of the research design, methods, and analysis.

3. The results are also not adequately presented. Figure 3 though interesting is much harder to read than a simple table. I believe further information is needed and climate change impacts need to be supported on further data.

To clarify the presentation of results, we included an additional table with the values in Appendix B - Table 3. We decided to keep the existing Figure 3 for three reasons: (1) It offers a way to visualize differences across survey respondents for specific and significant survey respondents; the maps inform the values per the six areas of Portland, which themselves geographically distinct and an important way to distinguish socio-economic differences (see preceding table); and (3) we expect the audiences of this paper -- urban forestry managers and scholars seeking new methods for measuring public perceptions of ecosystem services provided by trees -- to conduct similar studies for their regions. Figure 3 can also be interesting to support decision making by local community members who are familiar with the study area that might better assimilate relevant information. That said, if the Editors see this as a distracting figure, we can find alternative ways to provide relevant information through a narrative description.

 4. Conclusions should be written on a more scientific way. As they are they highlight the limitations of the research.

We understand that management of urban forestry using urban ecosystem services and socioeconomic indicators is not largely explored, and we endeavored to draw on our findings to support the extant literature. We have improved the conclusion to make them more scientific, by specifically reporting our findings and their implications for urban forest management.